# On the Theory of Magnetoelectric Coupling in Fe_2_Mo_3_O_8_

**DOI:** 10.3390/ma15228229

**Published:** 2022-11-19

**Authors:** Mikhail Eremin, Kirill Vasin, Alexey Nurmukhametov

**Affiliations:** Institute of Physics, Kazan Federal University, 420008 Kazan, Russia

**Keywords:** magnetoelecric coupling, ferroelectricity, crystal field

## Abstract

In the last decade, Fe_2_Mo_3_O_8_ was recognized for a giant magnetoelectric effect, the origin of which is still not clear. In the present paper, we contribute to the microscopic theory of the magnetoelectric coupling in this compound. Using crystal field theory and the molecular field approximation, we calculated the low-lying energy spectrum for iron ions and their interaction with electric and magnetic fields. Classical ionic contribution to the electric polarization related to the ionic shifts is also estimated. It is found that the electronic and ionic contributions to the electric polarization are comparable and these mechanisms support each other at $T<T_{N}$. The suggested electronic mechanism provides insight into the nature of huge jumps in polarization upon phase transitions from paramagnetic (PM) to antiferromagnetic (AFM) and then to ferrimagnetic (FRM) states under an applied external magnetic field as well as the large differential magnetoelectric coefficient.

## 1. Introduction

Fe_2_Mo_3_O_8_ crystals exhibit a record high polarization change upon the PM-AFM transition. In the antiferromagnetic phase, $P_{AF}$ is approximately 1400 μC/m${}^{2}$ [1,2] in absolute value. Moreover, the direction of the electric polarization vector coincides with the direction of the magnetic moments of the iron ions. This circumstance excludes the possibility of explaining the polarization on the basis of known magnetoelectric coupling mechanisms, such as the inverse Dzyaloshinskii–Moriya mechanism [3] or the spin current mechanism [4]. It is also incompatible with the hypothesis of the possible presence of toroidal moments [5], applicable to many other single-phase ME materials.

In [1], a mechanism of exchange striction was proposed to explain this induced polarization, i.e., it was assumed that when the antiferromagnetic ordering of the magnetic moments of iron is established, the equilibrium position of the lattice ions changes. The electric polarization vector of the sample was estimated as the sum of the products of the ion charges ($Q_{j}$) multiplied by the difference in the corresponding displacements of these ions along the *z* axis. The calculated polarization difference in the paramagnetic and antiferromagnetic phases turned out to be $\Delta P_{z}$ = 0.60(11) μC/cm${}^{2}$. This is about four times larger than the measured value, which, in our opinion, indicates an overestimated role of the ionic mechanism of polarization. Indeed, the displacement values used by the authors (which can be found in the appendix to the paper [1]) do not correspond to the recently obtained data on the Fe_2_Mo_3_O_8_ crystal structure in [6]. In the estimation of [1], ion displacements during phase transitions are selective. The displacements of bridging oxygen ions between the octahedral and tetrahedral positions of iron were assumed to be the largest. In the transition from the paramagnetic to the antiferromagnetic phase, they were considered to be approximately 0.04 Å. However, according to the X-ray and neutron diffraction data [6], the hypothesis of such selective oxygen displacements was not confirmed. The change in the structure with temperature decrease is described by the change in the lattice parameters. Moreover, during the phase transition at $T_{N}=60$ K, the lattice parameter a changes continuously. There is a slight kink in the change in the lattice parameter c, which corresponds to displacements of all lattice ions by about 0.002 Å, i.e., 20 times less than was assumed in [1].

The origin of magnetoelectric effects and the electric polarization jump were then discussed in [7]. A simplified theoretical model was used. Having calculated the energies of various phases, the authors did not confirm the conclusion of [1] that the experimentally observed features of the change in electric polarization during the transition from paramagnetic to antiferromagnetic and from antiferromagnetic to ferrimagnetic phases (provided a sufficiently strong external field is applied) can be explained only by the mechanism of ionic polarizability. The authors emphasized the need for a further more detailed study of the magnetoelectric effects in Fe_2_Mo_3_O_8_, taking into account the features of the electronic states of iron ions.

The Fe_2_Mo_3_O_8_ compound is a polar dielectric, that is, there is an electric polarization in the absence of an external electric field. As already noted in [1], it can be estimated from the formula $\Delta P_{z}=\frac{1}{V}\sum_{j}\left(z_{j}^{AF}-z_{j}^{PM}\right)Q_{j}$ using information about the positions of ions in the crystal lattice. We call this contribution ionic. In this paper, we pay attention to one more contribution to the electric polarization, which is induced by an external electric field. Iron ions in Fe_2_Mo_3_O_8_ are in positions without an inversion center and are subject to strong odd crystal fields. Odd crystal fields greatly enhance the polarizability of the 3d electrons of iron ions. We call this contribution to the total polarization of the crystal electronic.

The object of this paper is to study in detail the mechanisms of the electronic polarization of iron ions in Fe_2_Mo_3_O_8_. As mentioned in [7], this problem is complicated due to the orbital degeneracy of the ground multiplets of iron ions. The algorithm of our calculations is as follows. At the first stage, energy levels and wave functions of Fe${}^{2+}$ ions were calculated using modern crystal field theory methods. Then, effective operators of the interaction of 3d electrons with electric and magnetic fields were constructed in the basis of the lowest states of iron ions, taking into account the low-symmetry components of the crystal field and exchange (molecular) fields. At the final stage of the calculation, the free energy functions were determined, and then, graphs of the electric polarization of iron ions were plotted as functions of the external magnetic field and temperature.

## 2. Energy Levels Scheme of the Lowest-Lying Multiplets

In the Fe_2_Mo_3_O_8_crystal, iron ions occupy tetrahedral (A) and octahedral (B) positions, both have trigonal symmetry. With the quantization axis chosen along the third-order axis, crystal field operators can be written in the form:(1)$$H_{cf}=B_{0}^{\left(2\right)}C_{0}^{\left(2\right)}+B_{0}^{\left(4\right)}C_{0}^{\left(4\right)}+B_{3}^{\left(4\right)}C_{3}^{\left(4\right)}+B_{-3}^{\left(4\right)}C_{-3}^{\left(4\right)},$$
where $C_{q}^{\left(k\right)}$ are the components of the spherical tensor proportional to spherical functions
(2)$$C_{q}^{\left(k\right)}=\sqrt{\frac{4\pi }{2k+1}}\sum_{i}Y_{k,q}\left(\theta_{i},\phi_{i}\right).$$ The summation is carried out over all electrons in the 3d shell of Fe${}^{2+}$(3d${}^{6}$). The local coordinate axes at both positions can be chosen so that the imaginary parts of the crystal field parameters vanish. The eigenvalues of the operator (Equation 1) in the state basis of the ${}^{5}$D term are equal to
(3)$$\begin{array}{cc} \; & E_{4}=E_{5}-\frac{1}{14}B_{0}^{\left(2\right)}-\frac{1}{14}B_{0}^{\left(4\right)}+\frac{1}{2}{\left[{\left(\frac{5}{14}B_{0}^{\left(4\right)}-\frac{3}{7}B_{0}^{\left(2\right)}\right)}^{2}+\frac{5}{63}{\left|B_{3}^{\left(4\right)}\right|}^{2}\right]}^{1/2},\\ \; & E_{3}=\frac{2}{7}B_{0}^{\left(2\right)}+\frac{2}{7}B_{0}^{\left(4\right)},\\ \; & E_{1}=E_{2}=-\frac{1}{14}B_{0}^{\left(2\right)}-\frac{1}{14}B_{0}^{\left(4\right)}-\frac{1}{2}{\left[{\left(\frac{1}{14}B_{0}^{\left(4\right)}-\frac{3}{14}B_{0}^{\left(2\right)}\right)}^{2}+\frac{5}{63}{\left|B_{3}^{\left(4\right)}\right|}^{2}\right]}^{1/2}. \end{array}$$
The wave functions corresponding to these energy values are:(4)$$\begin{array}{cc} \; & |\psi_{1}\rangle =c_{1}|1\rangle -c_{2}|-2\rangle ,\\ \; & |\psi_{2}\rangle =c_{1}|-1\rangle +c_{2}|2\rangle ,\\ \; & |\psi_{3}\rangle =|M_{L}=0\rangle ,\\ \; & |\psi_{4}\rangle =c_{2}|1\rangle +c_{1}|-2\rangle ,\\ \; & |\psi_{5}\rangle =c_{2}|-1\rangle -c_{1}|2\rangle . \end{array}$$

In undistorted tetrahedral and octahedral coordinations, we have $B_{0}^{\left(2\right)}=0$ and $ReB_{3}^{\left(4\right)}=ReB_{-3}^{\left(4\right)}=\sqrt{\frac{10}{7}}B_{0}^{\left(4\right)}$, $c_{1}=\sqrt{2/3}$, $c_{2}=\sqrt{1/3}$. In this case, as follows from (Equation 3), $E_{4}=E_{5}=E_{3}$, i.e., we have an orbital triplet and an orbital doublet $E_{1}=E_{2}$. In the case of tetrahedral coordination, the ground state is the orbital doublet with five-fold degeneracy in spin variables ${}^{5}$E, and in the case of octahedral coordination, it is the orbital triplet ${}^{5}$T${}_{2}$. According to the spectroscopic data of [8], the energy interval between ${}^{5}$E and ${}^{5}$T${}_{2}$ multiplets (terms) for the A site is 0.5−0.6eV, and for the B site it is approximately 1.1−1.5eV. Using these data, we calculated the parameters $B_{q}^{\left(k\right)}$ on the Hartree–Fock wave functions of iron and oxygen ions, as was done in [9,10] for FeV${}_{2}$O${}_{4}$ and in FeCr${}_{2}$O${}_{4}$. In addition, they were corrected according to the available spectroscopic data given in [1,2,8] and according to the values of magnetic moments. The obtained parameters are given in Table 1.

The fine structure of ${}^{5}$E and ${}^{5}$T${}_{2}$ multiplets is formed due to spin–orbit interactions $H_{SO}^{\left(A\right)}=\lambda_{A}$LS, $H_{SO}^{\left(B\right)}=\lambda_{B}\mathrm{LS}$, exchange (molecular) fields $H_{EX}^{\left(A\right)}=I_{A}S_{z}$, $H_{EX}^{\left(B\right)}=I_{B}S_{z}$, as well as by the action of the spin–spin interaction ρ and the external magnetic field.

The magnetic moments measured using Mössbauer spectroscopy [11] are 4.21$\mu_{B}$ for the A site and $4.83\mu_{B}$ for the B site. They are mainly determined by the parameter $B_{0}^{\left(2\right)}$, which allows one to additionally refine and check the validity of the parameters from Table 1. The calculated energy levels are given in Table 2. The directions of the spins and the magnetic field are shown on Figure 1.

As can be seen from Table 2, the positions of the lowest-lying energy levels change significantly during the phase transition due to a change in the molecular field (parameter $I_{z}$), as well as due to the presence of an external magnetic field in the FRM phase. Correspondingly, the wave functions also change, which leads to a change in the matrix elements of the operator of interaction with the electric field.

The first column corresponds to the PM phase, the 2nd — to the AF phase, and the 3rd — to the FRM phase, transition to which is assumed to occur for *T*∼55 K and external magnetic field of ∼6 *T* (see Figure 1).

The value of the exchange (molecular) field was refined using the data of terahertz spectroscopy [2]. In the antiferromagnetic phase, we obtained $I_{A}≃I_{B}=72.5$ cm${}^{-1}$. As regards the exchange field in the FRM phase, the literature data are somewhat uncertain here. However, according to early works [7,11], it is possible to reveal a trend towards a decrease in the exchange fields at both sites by 2–3 times. To achieve the best fit with the experimental data, we used the values $I^{\prime }\left(B\right)≃$ 28 cm${}^{-1},I^{\prime }\left(A\right)≃I^{\prime }\left(B\right)$.

When the spins are directed along the trigonal symmetry axis (the **c** axis of the crystal), the wave function in the basis $|M_{S},M_{L}\rangle$ will take the form (the values of the coefficients are given in Table 3):(5)$$\begin{array}{c} |\psi_{g}\rangle =a|-2,-1\rangle +b|-2,2\rangle +c|-1,-2\rangle +d|-1,1\rangle +e|0,0\rangle +f|1,-1\rangle +g|1,2\rangle \\ +h|2,-2\rangle +i|2,1\rangle . \end{array}$$

## 3. Interaction of 3d-Electrons with Electromagnetic Radiation

The octahedral site of the iron ion is rather strongly distorted and, like the tetrahedral one, does not have an inversion center. The effective Hamiltonian of the interaction between 3d electrons and the electric field of light in both A and B sites has the form [12]: (6)$$H_{E}=\sum_{p,t,k}{\left\{E^{\left(1\right)}U^{\left(k\right)}\right\}}_{t}^{\left(p\right)}D_{t}^{(1k)p}.$$
Here, the curly brackets denote the direct (Kronecker) product of the spherical components of the electric field $E_{0}^{\left(1\right)}=E_{z}$, $E_{\pm 1}^{\left(1\right)}=\mp \left(E_{x}\pm iE_{y}\right)/\sqrt{2}$ and the unit tensor operator $U_{q}^{\left(k\right)}$. The summation indices take the values k=0,2,4, p=1,3,5, t=0,±3.

At the initial stage, the parameters $D_{t}^{(1k)p}$ were calculated in the same way as $B_{q}^{\left(k\right)}$ based on the Hartree–Fock functions according to the method described in detail in [9,10]. Then, they were corrected according to experimental data of terahertz spectroscopy [1]—Figure 2, electric polarization measurements [1], as well as the interference between electric and magnetic-dipole transitions in the related compound FeZnMo${}_{3}$O${}_{8}$ [13].

The operator of interaction with the magnetic field was written in the standard form:(7)$$H_{H}=\mu_{B}\left(L+2S\right)B.$$

Figure 2 shows the calculated absorption values for $I_{z}^{\left(A\right)}=72.5\;$cm${}^{-1}$ and parameters $D_{t}^{(1k)p}$ from Table 4. When calculating the absorption, both magnetic and electrical transitions were taken into account:(8)$$\kappa \propto \left|\langle i\right|H_{E}+H_{H}{\left|0\rangle \right|}^{2},$$
where the components of the incident light in the Gaussian units are related as $|H^{\omega }\left|=n\right|E^{\omega }|$ with *n* being the refractive index.

**Figure 2 materials-15-08229-f002:**
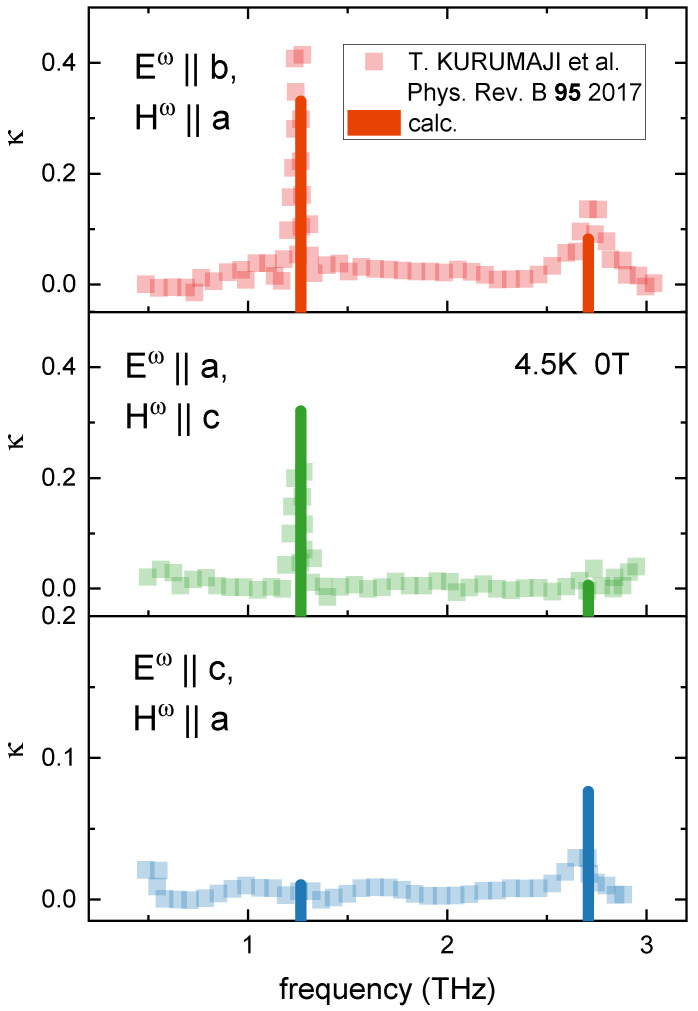
Calculated (vertical lines) and experimental (square symbols) [2] Fe_2_Mo_3_O_8_ absorption spectra in the THz region for various incident light polarizations.

An important conclusion about the relationship between the directions of magnetic moments and electric polarization can be made without specifying the values of the parameters of the spin–orbit coupling and the interaction of 3d electrons with electromagnetic fields. Recall that in previously suggested magnetoelectric coupling mechanisms [3,4], the electric polarization vector is always perpendicular to the direction of the spins. In this regard, it is interesting to understand how it turns out that the electric polarization in Fe_2_Mo_3_O_8_ is parallel to the magnetic moments [1,2]. The average energy of interaction with an electric field at low temperatures is determined by following the matrix element:(9)$$\begin{array}{c} \langle \psi_{g}\left|H_{E}\right|\psi_{g}\rangle =\left(a^{2}+f^{2}\right)\langle -1|H_{E}|-1\rangle +\left(b^{2}+g^{2}\right)\langle 2|H_{E}|2\rangle +\left(c^{2}+h^{2}\right)\langle -2|H_{E}|-2\rangle \\ +\left(d^{2}+i^{2}\right)\langle 1|H_{E}|1\rangle +e^{2}\langle 0|H_{E}|0\rangle . \end{array}$$
Here, we take into account that ${D_{t}^{(1k)p}}^{*}={\left(-1\right)}^{t}D_{-t}^{(1k)p}$ and $Im[D_{t}^{(1k)p}]=0$. All matrix elements $\langle M_{L}|H_{E}|M_{L}^{\prime }\rangle$ in this expression are proportional to the electric field component $E_{z}$. Matrix elements that depend on perpendicular components of the electric field vanish due to the selection rules. Consequently, the electric polarization induced by spin ordering is also directed along the **c** axis of the crystal, i.e., it is parallel to the direction of the spins. This conclusion is consistent with experimental data [1,2].

## 4. Dependence of the Electric Polarization of Iron Ions on the Magnitude and Direction of the External Magnetic Field and Temperature

Using the operator (Equation 6) on the wave functions obtained by diagonalizing the effective Hamiltonians, we have constructed electric polarization diagrams for both sites.

In the paramagnetic phase, the energy operator of iron ions contains the energy of 3d electrons in the crystal field, spin–spin and spin–orbit interactions. During the PM-AFM transition, the interaction energy of the spins of iron ions with the exchange (molecular) fields created by the surrounding spins is added. Applying the critical external magnetic field leads to the AFM-FRM transition and adds the Zeeman interaction to the Hamiltonian, as well as modifies the molecular fields. Accordingly, the energy level schemes and wave functions of iron ions change abruptly upon the phase transitions, which naturally affect the matrix elements of the operator (Equation 6).

The calculated temperature dependence of the electric polarization for individual positions of iron ions is shown in Figure 3. The temperature dependence is due to the change in the population of the lowest states of iron ions, the energies of which are given in Table 2. The electric polarization corresponding to the excited states differs from the polarization of the ground states. The total polarization was calculated in terms of the free energy function.

The magnetic field ($H_{c}$) dependence is shown in Figure 4. Note that the dependence on the magnetic field for individual ion positions contains both linear α and quadratic β contributions.

The graphs in Figure 4a,b are calculated for the positions A1 and B2, for the rest of the sites it is necessary to revert the directions of the molecular fields, or, what is equivalent, revert the signs of the α coefficients. Thus, the resulting magnetoelectric effect after averaging over all positions will be quadratic with respect to the magnetic field, as was experimentally found in [1].

### 4.1. Polarization Jump and Quasilinear Magnetoelectric Effect at the Boundary of the Phase Transition to the Ferrimagnetic State

For estimations, it is essential to take into account the reorientation of spins during the transition from AFM to FRM phases, caused by a change in the directions of the fields in the corresponding domains, as shown in Figure 1. Using the data from Figure 4, the electric polarization jump due to the electronic mechanism is estimated at −0.08μC/cm${}^{2}$ (experiment yields −0.13μC/cm${}^{2}$—see Figure 3b) in [1].

The origin of the electric polarization jump caused by iron ions is associated with a change in the symmetry of the AFM and FRM phases. As already noted in [2], the space symmetry group of the AFM phase is 6 mm${}^{\prime }$, while FRM is 6 m${}^{\prime }$m${}^{\prime }$. The linear magnetoelectric effect is possible for the 6m${}^{\prime }$m${}^{\prime }$ group, but is symmetry-forbidden in the antiferromagnetic phase. The microscopic calculation automatically takes into account the change in symmetry upon the phase transition also revealing new important details about the origin of the giant jump.

As one can see, the individual sites A and B in both phases already have a fairly strong linear magnetoelectric effect α; however, in the AFM phase, it is completely compensated by adding up all contributions from different positions in the unit cell, and only a much weaker quadratic effect remains visible. When the spins are reoriented to the FRM phase, the linear contributions no longer cancel each other, and, since the FRM phase exists only in a magnetic field, the polarization difference is already a large finite value. This is clearly seen in the section $H_{c}>6$ T [1]. Thus, the experimentally measured linear magnetoelectric effect is mostly associated with spin reorientation: the magnetic field modulation always occurs in a narrow region near the critical $H_{c}\left(T\right)$.

To clarify the physical nature, we have replaced the complex transient process with a step function and modeled the polarization behavior when the crystal is placed in a magnetic field modulated near its critical value. The results of our calculations are shown in Figure 5.

It is interesting to note that, following from our scenario, a change in temperature in the region of the critical magnetic field also changes the electric polarization. If we fix the amplitude of the alternating magnetic field $\Delta H_{c}\left(t\right)$ and lower the temperature (and with it the critical field, as follows from the phase diagram Figure 4c in [1])—we obtain a set of interesting graphs depicted in Figure 6.

In Figure 6a, the jump in electric polarization increases with decreasing temperature. Further, if we introduce the differential magnetoelectric coupling coefficient $\alpha_{diff}=dP_{c}{/dH|}_{H=H_{crit}\left(T\right)}$, where $H_{crit}$ is the critical field of the transition to the ferrimagnetic phase (see phase diagram in Figure 4c from [1]), we get the graph shown in Figure 6b. At present, only one experimental point corresponding to this graph is known. The origin of these effects can be explained as follows. Iron ions have several energy states split by the molecular field, which interact differently with the electric field, i.e., have different polarizations. These states are populated in the region of temperatures under consideration. When approaching low (helium) temperatures, only the ground states are populated; therefore, the change in the splitting values ceases to affect the change in the electric polarization $\alpha_{diff,A,B}\to 0$. In addition, according to the critical field of the transition to the ferrimagnetic phase, $H_{crit}$ also increases with decreasing temperature, increasing the initial splitting and suppressing the effect of the variable component of the external magnetic field even more. In the region of high temperatures, the population of the excited levels is so high that this leads to a significant compensation of the polarization of the iron ions. Thus, three competing phenomena lead to the appearance of an extremum in the differential coefficient $\alpha_{diff}$ of the magnetoelectric coupling. This theoretical prediction of the behavior of the magnetoelectric effect would be interesting to verify experimentally.

### 4.2. Ionic Mechanism of Electric Polarization

Let us make the rough estimate of the ionic contribution to the change in electric polarization upon the phase transition from the paramagnetic to the antiferromagnetic phase, using the crystal structure data from [6] for T=275 K and T=1.7 K. As in [1], we assume that the z-axis is parallel to the short bonds $R_{Fe-O}=1.946$ A in tetrahedral FeO${}_{4}$ fragments. Unit cell volumes are $V_{275K}=291.1$ A${}^{3},\;V_{1.7K}=290.661$ A${}^{3}$. The total dipole moment of ions in an electrically neutral system of charges does not depend on the choice of the origin. The summation is carried out over the ions of one unit cell: $\vec{p}=\sum_{i}q_{i}{\vec{r}}_{i}/V$. We get in μC/m${}^{2}$: ${\vec{p}}_{275K}=\left\{0,0,-137.744\right\}$, ${\vec{p}}_{1.7K}=\left\{0,0,-139.369\right\}$, polarization change between the antiferromagnetic and paramagnetic phases $\Delta p_{z}=-1.62484$ μC/m${}^{2}$.

This polarization change is comparable to the electronic mechanism contribution (see Figure 3 at T=1.7 K). Moreover, both contributions have the same sign, i.e., they support each other. Note that our estimate of the ionic contribution is probably overestimated, since it was obtained for a large range of temperature changes. As for the origin of the jump directly near the T=60 K region, it is logical to assume that it is mainly associated with the electronic mechanism, since there were no significant changes in the crystal structure at $T=T_{N}$ as it was reported in [6].

## 5. Concluding Remarks

We have studied in detail the origin of the magnetoelectric coupling related to iron ions. Using crystal field theory and molecular field approximation, we have calculated the low-lying energy spectrum for iron ions.

An effective operator for the interaction of 3d electrons of iron ions with an electric field was constructed, the parameters of which were refined by comparing the calculation results with experimental data on terahertz spectroscopy. This operator was then used to calculate the electronic polarizability due to iron ions. We explained the anomalously high polarizability of iron ions by a specific mechanism, which in some aspects is similar to that described in [9,10] for FeV${}_{2}$O${}_{4}$ and FeCr${}_{2}$O${}_{4}$. An important element of this mechanism is the presence of strong odd crystal fields, which admix the excited configurations of opposite parity to the main 3d${}^{6}$ electronic configurations.

In other words, the odd crystal field enhances the interaction of 3d electrons with an external electric field, and then, the spin–orbit interaction transfers the action of this enhanced electric field to the spin. The transfer of action to spins is carried out by processes of virtual excitations [9,10]. In Fe${}_{2}$Mo${}_{3}$O${}_{8}$, compared to FeV${}_{2}$O${}_{4}$ and FeCr${}_{2}$O${}_{4}$, the transfer efficiency is enhanced due to the presence of low-energy excited states. In addition, it should be taken into account that FeV${}_{2}$O${}_{4}$ and FeCr${}_{2}$O${}_{4}$ are noncollinear magnets. In this case, there is an additional mechanism associated with the exchange interaction of spins, which partially compensates for the first one. There is no such competing mechanism in Fe${}_{2}$Mo${}_{3}$O${}_{8}$, since it is a collinear magnet.

Generally speaking, there is one more source of electron polarization. It is related to the polarizability of oxygen and molybdenum ions. These ions, like iron ions, are in positions without an inversion center. They do not have magnetic moments, but may have induced electric dipole moments. However, the excited configurations of opposite parity for iron ions are much lower than for oxygen and molybdenum ions, so the role of the latter is expected to be relatively weak.

## Figures and Tables

**Figure 1 materials-15-08229-f001:**
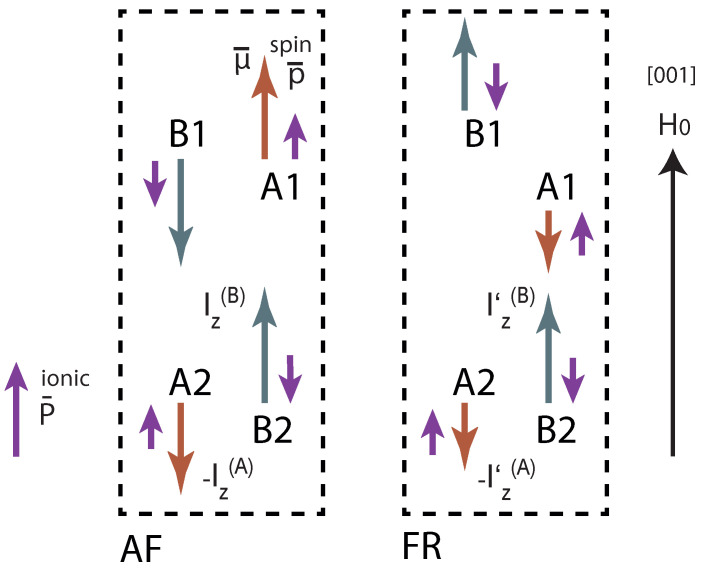
The direction of spins in each $Fe^{2+}$ sublattice (A—FeO${}_{4}$, B—FeO${}_{6}$) in the AFM and FRM phases, plotted according to [1]. Red (long) arrows indicate the direction of the magnetic moment μ of each sublattice. $I_{z}$, $I_{z}^{\prime }$—projections of molecular fields acting on iron ions in AFM and FRM phases, respectively. The short purple arrows show the directions of the calculated dipole moments. $P_{ionic}$—direction of the calculated ionic polarization vector of the lattice described in Section 4.2.

**Figure 3 materials-15-08229-f003:**
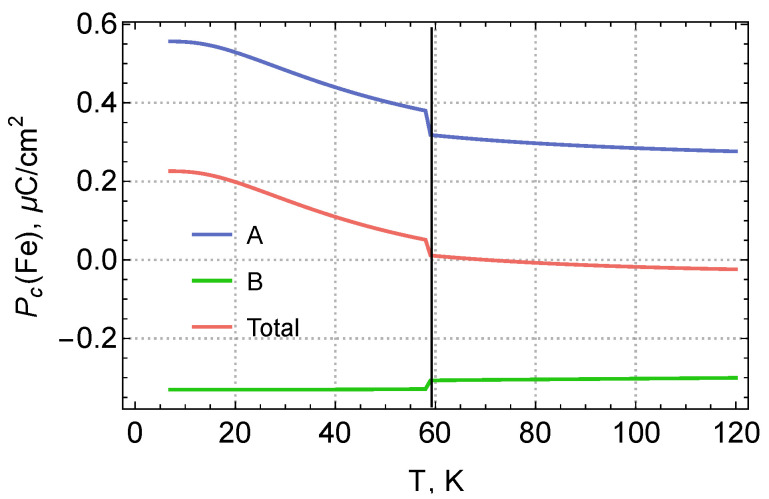
Calculated contributions to the electric polarization due to iron ions in the PM and AFM (T<60 K) phases.

**Figure 4 materials-15-08229-f004:**
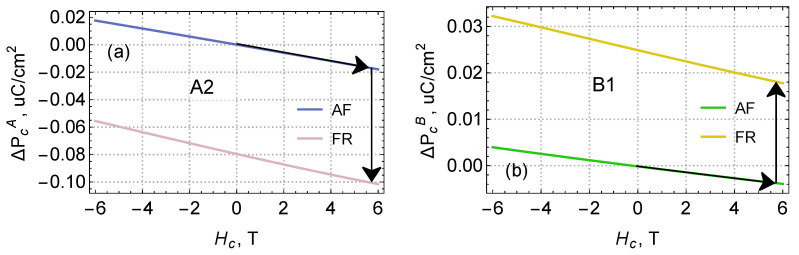
Field dependence of the electric polarization at positions A2 (**a**) and B1 (**b**) (see Figure 1) in the antiferromagnetic and ferrimagnetic phases. Calculated magnetoelectric coupling coefficients are: $\alpha_{AF}^{\left(A\right)}=-29.61$ μC/T, $\beta_{AF}^{\left(A\right)}=-0.06$ μC/T${}^{2}$, $\alpha_{FR}^{\left(A\right)}=-38.55$ μC/T, $\beta_{FR}^{\left(A\right)}=0.29$ μC/T${}^{2}$; $\alpha_{AF}^{\left(B\right)}=-6.57$ μC/T, $\beta_{AF}^{\left(B\right)}=0.05$ μC/T${}^{2}$, $\alpha_{FR}^{\left(B\right)}=-12.12$ μC/T, $\beta_{FR}^{\left(B\right)}=0.04$ $\mu C/T^{2}$ at temperature T=55 K. Black arrows demonstrate the path of the AFM-FRM phase transition.

**Figure 5 materials-15-08229-f005:**
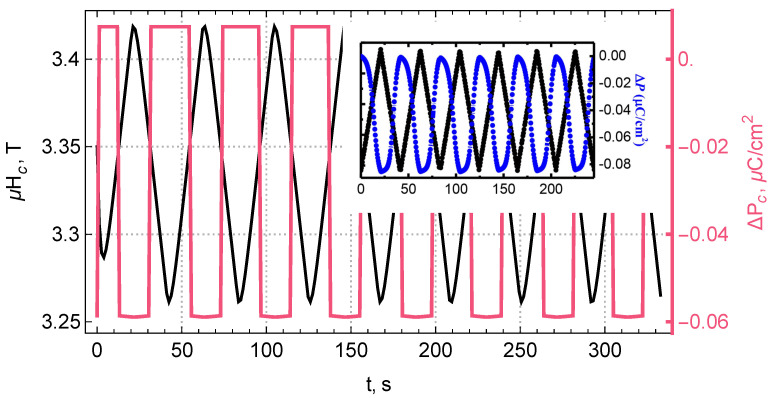
Modeled quasi–linear magnetoelectric effect at the AFM–FRM phase boundary. The magnetic field is modulated with time *t* by a triangular function with an amplitude of 0.2 T similar to the experiment near 3.25 T—the critical field of the transition to the ferrimagnetic phase. The inset shows the experimental plots from [1]—periodic modulation of electric polarization (blue) induced by a magnetic field (black) linearly varying between 3.25 T and 3.5 T at T∼58 K.

**Figure 6 materials-15-08229-f006:**
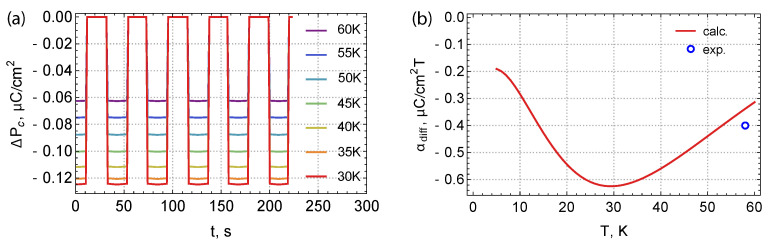
(**a**) The simulated quasi–linear magnetoelectric effect at the boundary between the antiferromagnetic and ferrimagnetic phases for various temperatures. The amplitude of the field oscillations is the same and corresponds to Figure 5. (**b**) Temperature dependence of the magnetoelectric coupling differential coefficient $\alpha_{diff}$. The experimental point on the graph corresponds to the work reported in [1].

**Table 1 materials-15-08229-t001:** Estimated parameters of energy operators in cm${}^{-1}$.

	$B_{0}^{\left(4\right)}$	$B_{3}^{\left(4\right)}$	$B_{0}^{\left(2\right)}$	λ	ρ	$c_{1}$	$c_{2}$
A	5500	6040	1250	−70	0.2	0.6241	−0.7814
B	−10,400	−11,500	1020	−100	0.2	0.8400	0.5427

**Table 2 materials-15-08229-t002:** The calculated energy levels. In each section, the 1st column corresponds to the energy splitting of the main term of $Fe^{2+}$ by the crystal field (CF) of a distorted tetrahedron/octahedron, and spin–orbit interaction (SO), 2nd and 3rd—estimated energy levels in the AFM and FRM correspondingly in the presence of the molecular field $I_{z}$.

A			B		
**PM**	**AFM**	**FRM**	**PM**	**AFM**	**FRM**
CF,SO	$\mathrm{CF},\mathrm{SO},I_{z}$	$\mathrm{CF},\mathrm{SO},I_{z}^{\prime }$	CF,SO	$\mathrm{CF},\mathrm{SO},I_{z}$	$\mathrm{CF},\mathrm{SO},I_{z}^{\prime }$
			507.847	790.545	700
			505.771	743.672	650
			505.771	735.605	650
			494.842	687.352	600
			494.842	636.856	600
50.7273	336.691	130	494.842	576.56	580
48.2967	289.273	100	410.423	479.506	490
36.6563	252.815	90	409.087	469.807	380
36.6563	229.168	80	242.932	343.325	300
23.9239	170.004	70	242.932	301.471	270
23.9239	169.74	70	168.221	293.118	250
15.7887	107.688	55	131.2860	288.564	230
5.7954	89.8808	55	34.6851	224.000	120
5.7954	42.1229	25	0	133.361	90
0	0	0	0	0	0

**Table 3 materials-15-08229-t003:** Coefficients for the ground state wave functions of iron ions in the antiferromagnetic phase. The wave functions for the opposite signs of the molecular field parameters I(A/B) can be obtained from those given here by the time sign reversal operation.

	a	b	c	d	e	f	g	h	i
A	0.785	0.543	0.219	−0.20	−0.01	−0.008	−0.006	0.000	−0.000
B	−0.007	−0.004	−0.244	−0.168	−0.413	−0.465	0.724	0.004	0.032

**Table 4 materials-15-08229-t004:** The calculated parameter values of the operator of interaction with the electric field (Equation 6) in a.u./|e|.

	$D_{0}^{(12)1}$	$D_{0}^{(12)3}$	$D_{0}^{(14)3}$	$D_{0}^{(14)5}$	$D_{3}^{(12)3}$	$D_{3}^{(14)3}$	$D_{3}^{(14)5}$
A	1.540	−2.177	5.849	−0.171	0.690	−3.403	0.086
B	−0.266	0.699	−1.234	−0.017	0.235	−0.240	0.063

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
