# Peer review of "On the Theory of Magnetoelectric Coupling in Fe2Mo3O8"

_materials, 2022, doi:10.3390/ma15228229_

Round 1

Reviewer 1 Report

In this paper, the authors used crystal field theory and molecular field approximation to calculate the low lying energy spectrum for iron ions and its interaction with electric and magnetic fields. However, there are a few questions the authors should address before it is suitable for publication.

1.       It is hard to understand that the electronic and ionic contributions to the electric polarization are comparable. Usually, the electronic contribution to polarization is negligible in comparison with the ionic contribution. How did you calculate the polarization contributed by ionic displacement? From the theoretical point of view, one may need the Born charge tensor to obtain the polarization or dipole moment. It seems that you were using a different approach to calculate polarization. Can you clarify that?

2.       In Figure 6(b), there is only one experimental data point. Can you add more points for a better comparison between experiment and calculation?

3.       There are a few typos in the text, please read it thoroughly and correct them.

Reviewer 2 Report

The work discusses intriguing topics related to multiferroic materials from a theoretical point of view. The theoretical approach is interesting to support material research, as it entitles anticipating specific physicochemical properties, e.g., Fe2Mo3O8 multiferroics.
The work is written clearly and in correct English. Therefore, it is recommended to publish in the journal Materials.

Author Response

Reviewer did not have any remarks. However, we believe that the improvements made according to other reviewers’ remarks made the methods description more clear.

Reviewer 3 Report

The English is very good, except for line 41 (toy) and line 46 (textbfasized). Please explain what you mean here.

Please explain why the electric polarization is always parallel to the magnetic moments in this case.

Please explain the electronic contribution to the polarization change.

Reviewer 4 Report

In this manuscript the authors reported the microscopic theory of the magnetoelectric 2 coupling in Fe2Mo3O8. This manuscript is well organized and the conclusion is well supported by the theory analysis. The current manuscript could be accepted after minor revision.

1) Fe2Mo3O8 is selected in this manuscript. Please illustrate the motivation by comparing with other single-phase ME materilas.

2) Can you make some comments about the potential value of the proposed theory in the field of single-phase multi-ferroic materilas. For example, can you propose some solutions to further improve the ME coupling capability accordingly?
